# TargetAntiAngio: A Sequence-Based Tool for the Prediction and Analysis of Anti-Angiogenic Peptides

**DOI:** 10.3390/ijms20122950

**Published:** 2019-06-17

**Authors:** Vishuda Laengsri, Chanin Nantasenamat, Nalini Schaduangrat, Pornlada Nuchnoi, Virapong Prachayasittikul, Watshara Shoombuatong

**Affiliations:** 1Department of Clinical Microscopy, Faculty of Medical Technology, Mahidol University, Bangkok 10700, Thailand; l.vishuda@gmail.com (V.L.); pornlada.nuc@mahidol.ac.th (P.N.); 2Center for Research and Innovation, Faculty of Medical Technology, Mahidol University, Bangkok 10700, Thailand; 3Center of Data Mining and Biomedical Informatics, Faculty of Medical Technology, Mahidol University, Bangkok 10700, Thailand; chanin.nan@mahidol.edu (C.N.); nalini.schaduangrat@gmail.com (N.S.); 4Department of Clinical Microbiology and Applied Technology, Faculty of Medical Technology, Mahidol University, Bangkok 10700, Thailand; virapong.pra@mahidol.ac.th

**Keywords:** anti-angiogenic peptide, therapeutic peptides, interpretable model, random forest, machine learning, classification

## Abstract

Cancer remains one of the major causes of death worldwide. Angiogenesis is crucial for the pathogenesis of various human diseases, especially solid tumors. The discovery of anti-angiogenic peptides is a promising therapeutic route for cancer treatment. Thus, reliably identifying anti-angiogenic peptides is extremely important for understanding their biophysical and biochemical properties that serve as the basis for the discovery of new anti-cancer drugs. This study aims to develop an efficient and interpretable computational model called TargetAntiAngio for predicting and characterizing anti-angiogenic peptides. TargetAntiAngio was developed using the random forest classifier in conjunction with various classes of peptide features. It was observed via an independent validation test that TargetAntiAngio can identify anti-angiogenic peptides with an average accuracy of 77.50% on an objective benchmark dataset. Comparisons demonstrated that TargetAntiAngio is superior to other existing methods. In addition, results revealed the following important characteristics of anti-angiogenic peptides: (i) disulfide bond forming Cys residues play an important role for inhibiting blood vessel proliferation; (ii) Cys located at the C-terminal domain can decrease endothelial formatting activity and suppress tumor growth; and (iii) Cyclic disulfide-rich peptides contribute to the inhibition of angiogenesis and cell migration, selectivity and stability. Finally, for the convenience of experimental scientists, the TargetAntiAngio web server was established and made freely available online.

## 1. Introduction

Cancer constitutes a group of diseases involving the unregulated proliferation of abnormal cells. It is capable of both invading surrounding normal tissue and spreading throughout the body via the circulatory or lymphatic system in a process known as metastasis [1]. Cancer is the second leading cause of death globally accounting for an estimated 9.6 million cases of death in 2018 [2]. However, early identification and treatment are able to increase the chances of survival for patients. As such, a combination of targeted drugs accompanied with chemotherapy or radiation is an essential strategy for ensuring an optimal outcome for patients [3,4].

Angiogenesis is a process by which new blood vessels are formed and it is seen as one of the key processes for the proliferation and metastatic spread of cancer cells. It promotes the circulation of oxygenated blood, supplies nutrients, and removes waste products from the body [5]. Furthermore, angiogenesis is regulated by both activator and inhibitor molecules with stimulation occurring when tumor tissues require oxygen and nutrients. However, the upregulation of the activity of angiogenic factors alone is not enough for the angiogenesis of neoplasm, as anti-angiogenic factors also needs to be downregulated [6]. 

Until now, various proteins have been identified and characterized as pro-angiogenic molecules including vascular endothelial growth factor (VEGF), transforming growth factor (TGF)-α, TGF-β, basic fibroblast growth factor (bFGF), angiogenin, and platelet-derived endothelial growth factor (PDGF) [7]. VEGF plays a central role in angiogenesis and is greatly expressed in cancer cells. Although there are many naturally occurring proteins that can inhibit angiogenesis (e.g., endostatin, angiostatin, platelet factor 4, and thrombospondin) [8], many researchers are still attempting to develop new anti-angiogenic molecules for inhibiting VEGF [9,10]. The conceptual basis for the discovery of novel anti-angiogenic molecules as a therapeutic route by means of VEGF inhibition is summarized in Figure 1. In addition, monoclonal anti-VEGF antibody (Avastin or Bevacizumab) is the first anti-angiogenic drug that is known for its ability to inhibit tumor blood vessel growth as well as its ability to increase survival rate in cancer patients [11,12]. However, blocking VEGF alone is not sufficient for stalling cancer cell growth and progression. Moreover, Sorafenib and Sunitinib [13], which are small molecules that can inhibit the VEGF receptor tyrosine kinase, has been approved for renal cell carcinoma and colorectal cancer treatments [14]. The production of various pro-angiogenic molecules besides VEGF for promoting tumor angiogenesis is a challenging endeavor, thus constructing new anti-angiogenic peptides represents an interesting avenue for novel therapeutics [15]. Lee et al. [16] reported a novel collagen IV derived biomimetic peptide that can inhibit breast cancer growth and metastasis. Its use in combination treatment with HER2 and VEGF peptides mimicked the induction of potent anti-tumor responses both in vitro and in vivo [17]. Therefore, the efficacy of anti-angiogenic peptides is dependent upon the cancer type. It is worthy to note that anti-angiogenic drugs are more efficient in well vascularized cancers. 

Apart from cancer, excessive vascular growth could promote blindness, rheumatoid arthritis and psoriasis [18]. Several studies have shown the effectiveness of anti-angiogenic peptides [15,19,20]. For example, luteolin has been demonstrated to be a potent anti-angiogenic agent for retinal neovascularization by suppressing the VEGF expression [21]. Additionally, anti-neuropilin-1 (anti-NP-1) was synthesized to block the function of NP-1, which is responsible for the induction of increased synoviocyte survival and angiogenesis. Thus, anti-NP-1 is useful in alleviating chronic arthritis [22]. Currently, peptide therapeutics are increasingly being used in medical practices against various diseases including cancers [15], microbial infections [23], and cardiovascular diseases [24]. Owing to several advantageous peptide properties such as high selectivity, efficacy, and being relatively safe, their use in the search for novel targeted drugs has gained much interest. However, several concerning factors should also be addressed, for instance, rapid degradation and excretion in humans, stability during storage, and low oral bioavailability [25]. In addition, most endogenous anti-angiogenic proteins are complex and too large, thereby causing them difficulty in penetrating target tissues. In this regard, therapeutic peptides were able to overcome these limitations with the development of anti-angiogenic peptides not exceeding 50 amino acids in length. Moreover, some of these peptides have been optimized and modified such as amino acid substitutions and conversion from L-to-D amino acids [26]. The effort of developing small peptide fragments that represent similar anti-angiogenic properties and could be applied to inhibit tumor angiogenesis in cancer patients represent valuable challenges. Besides, it is beneficial to construct specifically functional peptides as a result of an imbalance between activators and inhibitors that contributes to different pathological conditions. 

With the avalanche of post-genomic data, peptide sequences are abundantly available in databases. However, conventional experimental techniques for the identification and development of anti-angiogenic peptides have been very slow, expensive and laborious. Therefore, it is highly desirable to develop computational methods for predicting and characterizing anti-angiogenic peptides. However, until now, few efforts have been made to develop methods for accurately predicting anti-angiogenic peptides as summarized in Table 1. Ramaprasad et al. [27] first proposed a computational model named AntiAngioPred by using support vector machine in conjunction with amino acid composition and information on the first fifteen residues at the N-terminus region. This method gave accuracies of 75.00% and 74.96% as assessed by an independent validation test from 1 and 5 rounds of random splits, respectively. In 2018, Blanco et al. [28] utilized the three basic sequential features consisting of amino acid, dipeptide and tripeptide compositions for training and learning their prediction models. Their comparison results indicated that the best model based on generalized linear model yielded 86% accuracy in which such model utilized the top 200 informative features for model building. Recently, Zahiri et al. [29] developed the AntAngioCOOL R package and executed performance comparisons amongst various machine learning techniques and types of peptide features. Based on their results on performance comparisons, regression, and survival trees model employing descriptors consisting of pseudo amino acid composition, *k*-mer composition, *k*-mer composition (reduced alphabet), physico-chemical profile, and atomic profile yielded the highest accuracy of 77% over an independent validation test from 1 round of random split.

Although all these methods yielded encouraging results and played an important role in simulating the development of anti-angiogenic peptides identification, there is still room for further improvement in the prediction performance of anti-angiogenic peptides. The following research gaps have been elucidated as follows: (i) Blanco et al.’s method [28] and AntAngioCOOL [29] were assessed by the independent validation test from only 1 round of random split, hence, their prediction results were not yet satisfactory; (ii) the studies of Blanco et al.’s method [28] and AntAngioCOOL [29] did not provide a web server, hence, their usage was quite limited; and (iii) AntiAngioPred [27] and Blanco et al.’s method were not straight-forward enough to provide the underlying mechanism of anti-angiogenic peptides due of the lack of interpretability of the model.

Motivated by these considerations, this work attempts to develop a new sequence-based predictor for predicting and analyzing anti-angiogenic peptides, called the TargetAntiAngio, which utilizes the random forest classifier in cooperation with various types of peptide features including amino acid composition, dipeptide composition, physicochemical properties, pseudo amino acid composition, and amphiphilic pseudo amino acid composition. Rigorous cross-validation tests indicated that TargetAntiAngio outperformed the existing methods. Furthermore, this study also identified sequence features that contributed to high prediction accuracy as well as provided better understanding on the biophysical and biochemical properties of anti-angiogenic peptides by means of feature importance analysis. Finally, based on the proposed method, a user-friendly web server, called the TargetAntiAngio, was established for the prediction of anti-angiogenic peptides.

## 2. Results and Discussion

In this study, both 5-fold CV and independent validation test was performed on the benchmark ($S_{main}$) and NT15 ($S_{NT15}$) datasets. As mentioned in the section on the Benchmark dataset, the training and testing sets were constructed with the random sampling process. Furthermore, data splitting was performed with ten independent iterations to avoid the possible bias of the random sampling procedure. The final prediction results of the 5-fold CV and independent validation test were obtained by averaging the ten independent experiments. After which, comparisons of the prediction performances between the proposed method and the existing methods were conducted. Moreover, the informative features of AAC, DPC, and PCP were investigated to provide important biophysical and biochemical properties of anti-angiogenic activities of peptides. Finally, TargetAntiAngio was established as a free web server. Figure 2 shows the workflow of TargetAntiAngio which works in discriminating peptides as anti-angiogenic or non-antiangiogenic peptides. 

### 2.1. Prediction Performance

In order to predict and characterize anti-angiogenic peptides, it is very important to choose a useful classifier with informative features for the design of an accurate predictor as well as providing good understanding of anti-angiogenic activities of peptides. In this study, the five basic features (i.e., AAC, DPC, PCP, PseAAC, and Am-PseAAC) as well as their combinations (i.e., AAC+PseACC, AAC+Am-PseACC, PseACC+Am-PseACC, and AAC+PseACC+Am-PseACC) were selected as input features for training RF models followed identifying good combination of the five aforementioned features. 

Performance comparisons of the various feature types was performed for models built via 5-fold CV and independent validation test on the $S_{main}$ data set that was subjected to 1 random split and 10 rounds of random splits on the dataset as shown in Table 2 and Table 3, respectively. As noticed in Table 2, the highest test accuracy and MCC of 72.22% and 0.45, respectively, was achieved using the PseAAC feature. Meanwhile, the Am-PseAAC and ACC performed well with the second and third highest test accuracies of 72.22% and 72.12%, respectively. In order to yield better prediction performance, we also utilized the combinations of the top 3 important features (i.e., ACC, PseAAC and Am-PseAAC) to train the prediction models. The combination of PseACC and Am-PseACC reached a test accuracy and MCC of 77.78% and 0.56, respectively, while the combination of AAC and PseACC provided the second highest test accuracy and MCC of 75.93% and 0.52, respectively. In the case of the prediction results from 10 rounds of random splits, from amongst the top 3 important features, Table 3 shows that AAC had the best performance with a test accuracy and MCC of 73.33 ± 1.01% and 0.47 ± 0.02, respectively. Meanwhile, the combined features of AAC+PseACC and AAC+PseAAC+Am-PseAAC yielded the first and second highest test accuracy and MCC of 74.81 ± 1.01%/0.50 ± 0.02 and 74.07 ± 1.31%/0.49 ± 0.02, respectively. 

As mentioned in the section on the Benchmark dataset, it is not fair to compare our results with existing methods because AntiAngioPred was trained on the $S_{NT15}$ dataset. Therefore, in this study, the $S_{NT15}$ dataset was also utilized to develop the prediction models for comparative purposes. Performance comparisons of the RF models with various sequence features are summarized in Table 2 and Table 3. The highest test accuracy and MCC of 77.50 ± 1.77% and 0.56 ± 0.03 was achieved by using the combined features consisting of AAC, PseACC, and Am-PseACC. Meanwhile, the AAC feature and the combined feature of AAC+PseACC performed well as it afforded the second and third highest test accuracy and MCC of 77.00 ± 2.09%/0.55 ± 0.04 and 75.50 ± 1.12%/0.52 ± 0.02, respectively. As seen in Table 2 and Table 3, prediction results for the $S_{NT15}$ dataset are quite consistent with that of the $S_{main}$ dataset.

Furthermore, from Table 2 and Table 3, the experimental results can be briefly summarized hereafter. Each of the three single features including AAC, PseACC, and Am-PseACC are benefical for predicting anti-angiogenic peptides with test accuracies of >73% and >77% when performed on $S_{main}$ and $S_{NT15}$ datasets, respectively. Furthermore, prediction results for the $S_{NT15}$ dataset were better than that of the $S_{main}$ dataset thereby indicating that the position of the first fifteen residues plays a vital role in discriminating anti-angiogenic from non-antiangiogenic peptides (Table 2 and Table 3). This observation is in good consistency with the study of Ramaprasad et al. [27]. The best prediction performance for both $S_{main}$ and $S_{NT15}$ datasets as evaluated via independent validation test from 10 rounds of random splits were achieved by using the combined features of AAC, PseACC, and Am-PseACC. For convenience, we will refer to this RF method built with the combined feature of AAC, PseACC, and Am-PseACC as TargetAntiAngio.

### 2.2. Comparison with Other Methods

It is necessary to compare our proposed method TargetAntiAngio with that of the existing methods by performing both cross-validation and independent validation tests so as to ascertain its efficiency and strength for the prediction of anti-angiogenic peptides. Until now, there are only three sequence-based predictors that have been developed for identifying anti-angiogenic peptides consisting of AntiAngioPred [27], Blanco et al.’s method [28], and AntAngioCOOL, as summarized in Table 1. From amongst these three predictors, only AntiAngioPred provided prediction results that are rigorously assessed by both cross-validation and independent validation tests as assessed by more than 1 round of random split (i.e., evaluated on both the $S_{main}$ and $S_{NT15}$ datasets). In view of this point, herein, we only compared our method TargetAntiAngio with AntiAngioPred [27]. Table 4 lists the performance comparisons between TargetAntiAngio and AntiAngioPred over 5-fold cross-validation and independent validation tests using the $S_{NT15}$ dataset.

Based on the results from Table 4, it can be seen that TargetAntiAngio afforded a lower performance as compared to AntiAngioPred (75.00% vs 80.90% accuracy) as assessed by 5-fold cross-validation from one round of random split. On the other hand, prediction results for models evaluated by the independent validation test using dataset obtained from one round of random split indicated that TargetAntiAngio achieved better performances than that of AntiAngioPred as observed from the values of accuracy (77.50% vs 75.00%) and MCC (0.56 vs. 0.51). Furthermore, TargetAntiAngio was also found to outperform AntiAngioPred with improvements of 3% and 6% for both accuracy and MCC, respectively, as evaluated by a robust independent validation test using datasets obtained from 10 rounds of random splits.

### 2.3. Biological Space

The analysis of feature importance can provide a better understanding of the mechanistic details governing the anti-angiogenic activity of peptides. As mentioned above, in this study, the informative features of AAC, DPC, and PCP were used to characterize the anti-angiogenic activity of peptides. In order to select informative features, this study utilized the RF model because of its built-in ability of feature importance estimation and its great prediction performance. The value of mean decrease of Gini index (MDGI) is adopted to rank and estimate the importance of each AAC and DPC features. Such information is derived from analysis of the $S_{main}$ dataset that consists of 137 anti-angiogenic and 137 non-antiangiogenic peptides. Table 5 lists the percentage values of the 20 amino acids for both anti-angiogenic and non-anti-angiogenic along with their amino acid compositional difference between the two classes along with their MDGI values. Features with the highest MDGI value is considered as the most important as it significantly contributed to the prediction performance. As seen in Table 5, the 10 top-ranked informative amino acids with the highest MDGI values are Cys, Ser, Val, Ala, Leu, Arg, Glu, Lys, and Pro afforded MDGI values of 15.90, 14.43, 9.58, 9.21, 8.41, 8.31, 6.68, 6.59, 6.40, and 5.52, respectively. Meanwhile, from amongst the 10 informative amino acids, the analysis of AAC with the percentage of certain residues on anti-angiogenic peptides suggested that Cys, Ser, Arg, and Pro are dominant in anti-angiogenic peptides, while Val, Ala, Leu, Glu, and Ile are dominant in non-antiangiogenic peptides at the significance level of *p*-value ≤ 0.05. 

Furthermore, the sequence logo of the first and last fifteen residues at the N- and C-terminal regions of both anti-angiogenic and non-antiangiogenic peptides were created to visualize the positional information for each amino acid as shown in Figure 3. The overall stack height of each position indicates its sequence conservation while the size of the residue represents its propensity. Figure 3a,c shows that Pro, Ser, Trp, Cys, and Gly as well as Cys, Ser, Gly, Pro, and Arg are abundant at the first 15 residues from the N- and C-terminal regions, respectively, of anti-angiogenic peptides. However, only Leu and Ala are abundant at the last 15 residues from the C-terminal region of non-antiangiogenic peptides. Thus, information gathered from the sequence logo illustration shows crucial amino acid residues that could potentially be used for discriminating anti-angiogenic from non-antiangiogenic peptides. Moreover, Cys, Ser, and Arg are seen to be favored by anti-angiogenic peptides, especially at the C-terminal region. These analyses were in good consistency with the feature importance as estimated using MDGI values where Cys, Ser, and Arg are ranked 1, 2, and 6, respectively (Table 6). 

The heatmap of feature importance for the DPC feature can be seen in Figure 4, from which, the 20 top-ranked informative dipeptides with the highest MDGI values are SP, TC, CG, CS, SC, TR, RT, PF, AS, HG, LI, PC, RP, AA, SL, AL, ST, IV, RR, and AD. From amongst the top 20 informative dipeptides, there are 6 dipeptides (SP, TC, CG, CS, SC, and TR) with MDGI values larger than 1.45. In addition, 4 out of the 6 top-ranked informative dipeptides (TC, CG, CS, and SC) consist of Cys, while 3 out of the 20 top-ranked informative dipeptides (TR, RT, and RP) consist of Arg. As mentioned previously, Cys and Arg were the first and sixth important amino acids with the highest MDGI values of 15.90 and 8.31, respectively. These results reinforced the importance of Cys and Arg for the anti-angiogenic activity of peptides. Furthermore, detailed analysis of these two amino acids are discussed below.

Cys provided the largest MDGI value and results shown in Table 5 displayed that the percentage composition of Cys residues are found to be significantly different in a comparison between anti-angiogenic (0.047%) and non-antiangiogenic (0.014%) peptides producing significant p-value < 0.05. Many studies have reported that Cys is the preferred residue for anti-angiogenic activity [30,31,32,33]. Cys is classified as a polar, non-charged amino acid containing sulfur which, when oxidized, could form a disulfide bond. It stabilizes the tridimensional structure, which is essential for extracellular proteins that might be exposed to virulent conditions. Peptides containing multiple disulfide bridges are more resistant to thermal denaturation and is also crucial for maintaining their biological activity [34]. In 1997, a globular protein namely, endostatin was first discovered by Folkman and coworkers as an endogenous inhibitor of angiogenesis [35]. Mass spectrometry demonstrated that endostatin contains two disulfide bonds: Cys162-302 and Cys264-294 [31,32]. In addition, histological sections of tumors from saline-versus-endostatin-treated Lewis lung carcinomas were analyzed for apoptosis and angiogenesis. The results showed that the apoptotic index of tumor cells increased 7-folds (*p*-value < 0.001) while angiogenesis was completely suppressed in tumor cells (*p*-value < 0.001) for the endostatin treated mice [35]. Furthermore, Hiraki et al. [36] performed site-directed mutagenesis of chondromodulin-1 (ChM-1) as to assess the importance of Cys toward the function of ChM-1. The results disclosed that the ChM-1 mutant, which had all eight Cys residues replaced by Ser, lost the inhibitory effect of VEGF-A that subsequently stimulated the migration of human umbilical vein endothelial cells (HUVEC) due to the lack of disulfide bonds. Remarkably, Ser at positions 83 and 99 on the replaced ChM-1, revealed a decreased cell migration (150%) as compared to that of VEGF-A (350%). This result indicated that the disruption of one disulfide bond cannot neutralize its migratory effect. In addition, the Δ (Cys83 Cys99) rhChM-1 mutant lacking the 17 amino acid residues from Cys^83^-Cys^99^ and but retained three disulfide bonds, still appeared to exhibit its inhibitory effect [37]. Similarly, Chlenski et al. [20] designed and synthesized two peptides consisting of FSEC (CELDENNTPMC) and FSEN (CQNHAKHGKVC) from FS-E (CQNHCKHGKVCELDENNTPMC) by linking Cys 1 to Cys 3 and Cys 2 to Cys 4, owing to the need to construct simpler peptides with less complex structures. FS-E is classified in the group of secreted protein acidic and rich in cysteine (SPARC). In this study [20], the authors divided the experimental processes into three parts including: (i) endothelial cell migration assay (ii) inhibition of neuroblastoma tumor growth and (iii) inhibition of tumor induced angiogenesis. Firstly, in order to evaluate the capability of the two simple peptides to inhibit endothelial cell migration, HUVEC were treated with serial dilution of FSEC and FSEN by monitoring the percentage of stimulation compared with beta-fibroblast growth factor (bFGF) as a positive control. For the former, in vitro experiment was demonstrated by the inhibition of human umbilical vein endothelial cells (HUVECs) migration with an EC_50_ of 1 pM. Secondly, an in vivo experiment was demonstrated via a mice model in which mice with subcutaneous neuroblastoma xenografts were treated with the FSEC peptide for 2 weeks. The FSEC-treated mice were compared to the control group (PBS) and it was revealed that the inhibition of tumor growth was observed as deduced from the decreasing tumor weight (*p* = 0.01). Lastly, a paraffin section of xenografted mice was stained using green CD31 (PCAM-1) positive endothelial cells and red SMA-positive pericytes whereby the quantity of tumor blood vessels was calculated as the area occupied of staining. Results revealed that FSEC was significantly reduced in FSEC treated xenografts as compared to vesicle treated control (p-value < 0.001). This study also indicated that FSEC, which is a modified linear peptide containing disulfide bonds, has the ability to completely abrogate angiogenesis thereby leading to tumor growth inhibition. Their results is consistent with previous studies that SPARC can inhibit breast cancer progression [38], ovarian metastasis [33] with the overexpression of endogenous angiogenic inhibitors such as somatostatin, angiostatin, and endostatin, which also represents negative correlation with poor prognosis of cancer patients [39,40].

Furthermore, Yang X et al. [41] modified wild-type (WT) kringle5 (K5), which has been shown to contain anti-angiogenic activity with higher potential than angiostatin, by disruption of its disulfide bond distribution. K5mut1 was designed by deleting amino acid residues outside the kringle domain whereas Cys462-Cys451 is still located in the WT K5. Additionally, K5mut2 was constructed by removing Cys462, thereby leading to the loss of one disulfide bond. The effect of WT K5 and its deleted mutation on endothelial cell proliferation, cell apoptosis, and tumor growth were evaluated by the percentage of cell viability, flow cytometry and tumor weight, respectively. In vitro results showed that K5mut1 was able to decrease endothelial cell proliferation by 2-fold, enhancing endothelial cell apoptosis. Moreover, in vivo experiment was revealed that the weight of liver tumor in a mouse model was gradually decreased compared to mice treated with wild-type K5. Meanwhile, K5mut2 lacking one Cys, lost all its inhibitory effects. In summary, anti-angiogenic peptides containing Cys residues that formed disulfide bonds play an important role in (i) inhibiting blood vessel proliferation through the activation of angiostatin contributes to a lack of nutrients and blood supply to tumor cells [20,42], (ii) increasing anti-angiogenesis via reduction of specific receptors for pro-angiogenic molecules, (iii) inducing cell apoptosis [35,43], and (iv) balancing opposing signals in the tumor microenvironment [44].

Although our prediction model showed that Cys is the most important amino acid for the inhibition of blood vessel proliferation and tumor growth, other peptides which does not contain Cys have also demonstrated anti-angiogenic activity. Recent advances in biotechnology have led to the discovery of numerous biologically active peptides. The challenge is to also increase other physicochemical properties such as the bioavailability and as such pharmaceutical techniques such as liposome, hydrogel, nanoparticle, and targeted drug delivery system should be utilized for improvement of the potency of anti-angiogenic peptides. For example, tumstatin peptides binds to avB3 integrin on proliferating endothelial cells and also localizes to the target tumor. Moreover, when combined with bevacizumab (anti-VEGF antibody), an increase in its efficacy against tumor progression was observed [45]. Thus, the design of therapeutic peptides utilizes appropriate amino acids for bringing about the intended effect as to target specific mechanisms of interest. Representing the sixth largest MDGI value (Table 5), the percentage composition of Arg residues is found to be significantly different between anti-angiogenic (0.088%) and non-antiangiogenic (0.055%) peptides at a significance level of p-value < 0.05. Bae et al. [46] identified hexapeptides from peptide libraries in order to investigate their effects on the binding of VEGF to their receptors. The authors found that the most important amino acids for inhibitory activity included Arg, Lys, and His. Meanwhile, three peptides RRKRRR, RKKRKR, and hexa-arginine (RRRRRR) were demonstrated to be the most effective inhibitor with IC_50_ values of 2, 3.4, and 3.8 μM, respectively. In addition, the interaction between hexapeptides and VEGF was investigated by monitoring the binding of labeled VEGF_165_ to endothelial cells. Results showed that Arg-rich (AR) hexapeptides directly binds to VEGF_165_ (K_D_ = 5, 2 and 22 μM). Furthermore, the proliferation assay also confirmed that AR hexapeptides inhibited HUVE cell proliferation by VEGF_165_ in a concentration-dependent manner without cytotoxicity. Moreover, the essential role of hexapeptides containing basic charged amino acid resides was elucidated via blocking the metastasis of human colon carcinoma cells. Results disclosed that RRKRRR decreased the number of metastatic nodules by 16% as compared to that of the control whereas hexa-Lys (KKKKKK) showed minor inhibitory effects (80% of control). Conversely, the peptide with negative charge (EEFDDA) appeared to show no inhibitory activity at all. In addition, Xiong et al. [47] demonstrated that treatment of cells with 0.05 mmol/L of L-Arg for 7 days caused endothelial dysfunction as measured by the enhanced superoxide anion and decreased NO production. Thus, the chronic L-Arg supplementation is potent for accelerating endothelial cell senescence expression with the up-regulation of Arg-II. Moreover, Arg was utilized to create a synthetic RGD (Arg-Gly-Asp) integrin ligand sequence for improving the tumor cell targeting capability of therapeutic peptides [48]. Xu et al. [49] synthesized HM3 peptide (IRRADRAAVPGGGG) and added RGD (IRRADRAAVPGGGG-RGD) in their investigation on the inhibitory effect. The experimental result showed that it could significantly inhibit the migration of the HM3 peptide into endothelial cells. Besides, Matrigel and aortic ring tests conducted in a mice model also revealed that HM3 could potentially inhibit angiogenesis. Similarly, Buerkle et al. [50] explored the effect of cyclic RGD peptide as an α_v_-integrin antagonist on angiogenesis, microcirculation, growth, and metastatic formation of solid tumors. Results indicated that the cyclic RDG peptide reduced blood vessel density as well as diminished tumor growth and metastasis. Additionally, Kando et al. [51] developed a liposomal drug targeted to membrane type-1 matrix metalloproteinase by modification with stearoyl Gly-Pro-Leu-Pro-Leu-Arg (GPLPLR). The authors observed that the modified liposome showed high binding ability to HUVEC and increased accumulation in tumor cell (> 4-fold). In summary, peptides containing Arg induced anti-angiogenic activity and contributed to the inhibition of tumor growth via (i) the binding of peptides to the main body of VEGF including the N- and C-terminal ends [46] and (ii) increasing the specificity to targeted tumors as Arg confers a small positive charge there allowing cell binding via electrostatic interactions with the negatively charged cell membranes thus, leading to arrested tumor growth [52]. 

### 2.4. Mechanistic Interpretation of Informative PCP

Physicochemical properties of amino acids play an essential role as effective features for identifying and characterizing the functions of protein or peptide from their primary sequences [53,54,55,56,57]. It is well known that PCP [58], such as molecular volume, exposure or accessible surface, polarity (hydrophobicity/hydrophilicity), charge/pK, hydrogen-bonding potential and so forth are correlated with the structure and function of the amino acid sequence [59]. Herein, we have obtained the 10 top-ranked informative PCPs corresponding to their MDGI values, as shown in Table 6. As seen in Table 6, CHOP780215, CHOP780214, and CHOP780213 represents the second, third, and fourth important PCPs with corresponding MDGI values of 0.61, 0.54, and 0.50, respectively. Meanwhile, another important PCP is CHOP780209 with a corresponding MDGI value of 0.34 was not found in the top 10 informative PCPs. The four important PCPs of anti-angiogenic peptides were analyzed and discussed below.

#### 2.4.1. Peptides Having Cys Locating at the C-terminal Domain Can Decrease Endothelial Formation Activity and Suppress Tumor Growth

CHOP780209 with a corresponding MDGI value of 0.34 is described as the normalized frequency of the C-terminal β-sheet. The secondary structure prediction by the Chou-Fasman method demonstrated that the conformational preferences of Cys in adopting the β-strand structure is 1.19 [60]. It is well-known that the β-strand is a stretch of polypeptide chain containing approximately 3–10 residues in length. The interaction among more than two β-strands (around six β-strands) via hydrogen bonds could form the β-sheet structure [61]. Based on this PCP, it could be stated that the β-sheet structure and Cys residue are important for anti-angiogenic activities of peptides. Previous studies reported that endostatin [62,63], thrombospondin (TSP) [64,65], somatostatin [66], ChM-I [67], and TeM [65] containing Cys rich domain at the C-terminal region is likely to adopt the β-sheet structure. Figure 5 shows the three-dimensional structures of endostatin (a), somatostatin (b) and Platelet factor-4 (c). Moreover, an endogenous angiogenic inhibitor was revealed for controlling angiogenic balance. Hohenester et al. [63] reported that the crystal structure of endogenous angiogenic inhibitor namely endostatin, is a fragment derived from the C-terminal domain of collagen XVIII, containing two disulfide bridges located in the β-sheet [63]. After blocking angiogenesis, it is accompanied by high proliferation that is balanced by apoptosis in tumor cells [35]. In addition, Talaboletti et al. [64] showed that TSP has the ability to inhibit tumor cells migration. TSP-1 is an essential fragment of TSP at the C-terminal region that includes four Cys residues (two intra disulfide bonds). TSP-1 binds to CD36 receptors on endothelial cells that could allow for endothelial-cell apoptosis thereby leading to the inhibition of angiogenesis [65]. Furthermore, Ginj et al. [66] synthesized and evaluated the biological activity of somatostatin-based radiopeptides. The authors found that the peptides improved the binding affinity toward tumor cells and enhanced the internalization into cells expressing somatostatin receptor [44]. Hiraki et al. [36] revealed that mature human chondromodulin-I (ChM-I) consists of 120 amino acids and the C-terminal hydrophobic domain (Phe42 to Val120) in the *β*-sheet region indicates a functional domain for the inhibition of vascular endothelial cell growth in vitro. In order to present the structural requirements for ChM-I to exert its anti-angiogenic activity, Miura et al. [37] observed that the C-terminal domain containing eight Cys residues successfully inhibited cell migration with the decrease in percentage of migrated cells (200%) as compared to VEGF-A (>400%). Additionally, Oshima et al. [67] reported that the linear peptide, Tenomodulin (TeM), has the potential to be an anti-angiogenic peptide in vitro and in vivo. TeM is a well-known cartilage-derived angiogenesis inhibitor containing eight cysteine residues and a unique disulfide bridged hydrophobic domain at the C-terminal region, which adopts a β-sheet structure. The authors confirmed the functional role of Cys at the C-terminal region of TeM that are essential for the anti-angiogenic activity by monitoring the Matrigel tube formation and measuring the tube length. They further cleaved and constructed a secreted C-terminal domain of TeM (shTEM) containing the Cys-rich domain from human TeM. As in vitro result, their experiment showed that shTeM had more potential to restrain HUVEC cells at a low concentration of 50 µM when compared to the MOCK control (220 µM). In addition, they demonstrated in vivo experimental result that shTeM transduced melanoma cells formed tumors in C57BL/6 mice that were 46% smaller than enhanced green fluorescent protein (EGFP) transduction. In summary, Cys present at the C-terminal domain is one crucial factor influencing synthetic anti-angiogenic peptides in order to decrease endothelial formation activity and suppress tumor growth.

#### 2.4.2. Cyclic Disulfide-Rich Peptides Provide Greater Inhibition of Angiogenesis and Cell Migration, Selectivity, and Stability than Linear Peptides

The second, third, and fourth important PCPs (CHOP780215, CHOP780214, and CHOP780213, respectively) describes the turn of a protein structure. β-turn is thought to be involved in protein folding initiation while its conformational structure also contributes to protein stability and the free energy of proteins [68]. Moreover, the Chou-Fasman method of secondary structure prediction showed that the conformational preferences of Cys to be β-turn is 1.19 [60]. Many studies have reported that the increase in stability of protein structures as afforded by the β-turn is represented by disulfide bridges of the CXXC motif that is formed when the peptide brings together the first and fourth Cys to form a cyclic disulfide bond [69,70]. Although disulfide bonds located in linear peptides demonstrated anti-angiogenic activity, however, many researchers used the cyclic modified peptide conformations in order to promote their stability. Accordingly, cyclic peptides were able to exhibit enhanced cell permeability, increased bioavailability, and binding specificity [71]. Miura et al. [37] revealed the structural requirements of ChM-1 for enhancing its anti-angiogenic property by observing its prevention of the VEGF-A induced migration of HUVEC. The result showed that the synthetic ChM-1 cyclic peptide linked by disulfide bond between Cys^83^ and Cys^99^ promoted a migratory effect via a dose response curve (ID_50_ value of 2 µM) as compared to the ChM-1 linear peptide. In addition, the ChM-1 cyclic peptide was modified at the hydrophobic C-terminal tail and then examined for its effects on tumor angiogenesis. The authors revealed that the tailed ChM-1 cyclic peptide clearly decreased the tumor volume. Similarly, researchers [72,73] have stated that cyclic disulfide-rich peptides, including six inter-cysteine loops (namely *Momordica cochinchinensis* trypsin inhibitor-II (MCoTI-II)), have a high enzymatic and thermal stability. Thus, Chan et al. [19] designed a second-generation grafted cyclic peptide (MCoAA-02) combined with anti-angiogenic epitopes (somatostatin (SST)-01 and pigment epithelium-derived factor (PEDF)) [74]. The authors observed a high potency of 50% inhibition of HUVEC migration at 1 nM using the MCoAA-02 peptide. Based on its potency, MCoAA-02 was evaluated using the chorioallantoic membrane (CAM) assay to monitor its effects on blood vessel growth in vivo. Results revealed that MCoAA-02 had comparable capability to that of cyclic peptide-based drugs namely, octreotide at 10 µM. In addition, MCoAA-02 also exhibited high stability as compared to orally active sunitimib as observed through the percentage of peptide remaining over 24 h in a serum stability assay [19]. In conclusion, re-engineering linear peptides into cyclic peptides resulted in enhancing the inhibition of angiogenesis, cell migration, selectivity, and stability. Furthermore, the combination between cyclic peptides and potent anti-angiogenic agents provided a synergistic effect which poses an opportunity for the exploration of better therapeutic peptides [75,76,77,78].

### 2.5. TargetAntiAngio Web Server

To afford wide utilization of the prediction capability of the QSAR model, we had constructed a web server called the *TargetAntiAngio*. The web interface was established using the *Shiny* package under the R programming environment. The web server is freely accessible at http://codes.bio/targetantiangio/ (accessed on: 1 April 2019). Screenshots of the TargetAntiAngio web server is shown in Figure 6. A step-by-step guide on using the web server to get the desired results is given below:Step 1. Open the web server TargetAntiAngio at http://codes.bio/targetantiangio/ (accessed on: day 1 April 2019)Step 2. Either enter the query sequence into the Input box or upload the sequence file by clicking on the “Choose file” button (i.e., found below “the Enter your input sequence(s) in FASTA format heading”).Step 3. Press on the “Submit” button to initiate the prediction process.Step 4. Prediction results are automatically displayed in the grey box found below the “Status/Output” heading. Typically, it takes a few seconds for the server to process the task. Users can also download the prediction results as a CSV file by pressing on the “Download CSV button”.

Additionally, users could also run a local copy of TargetAntiAngio on their own computer using a one-line code as follows in an R environment: 

shiny::runGitHub(‘targetantiangio’, ‘chaninlab’, subdir = “ targetantiangio_shiny_server “)

However, prior to running the aforementioned code, it is recommended that users first install the prerequisite R packages. This can be performed by using the following code: 

install.packages(c(‘shiny’,  ‘shinyjs’,  ‘shinythemes’,  ‘protr’,  ‘seqinr’,  ‘caret’,  ‘markdown’)).

## 3. Materials and Methods

### 3.1. Benchmark Dataset

The first and most important consideration for developing a promising computational model is to construct a reliable benchmark dataset. In this study, the benchmark dataset was obtained from the work of Ramaprasad et al. [27], which has been used for developing recent prediction models of anti-angiogenic peptides [28,29]. Initially, the benchmark dataset had 257 peptide sequences in the anti-angiogenic class as derived from various articles and patents. To obtain a good quality benchmark dataset, the following steps were considered. Firstly, to avoid a dataset containing many redundant peptides, anti-angiogenic peptides having >90% of sequence similarity was filtered out using the CD-HIT program [79]. Secondly, anti-angiogenic peptides containing special characters, such as X and U, were removed. After such screening procedures, a set of 135 peptide sequences belonging to the anti-angiogenic class was obtained. Due to the lack of peptide sequences for the non-antiangiogenic class, 135 random peptides were used as non-antiangiogenic peptides. The benchmark dataset ($S_{main}$) used in this study can be summarized by the following formula:(1)$$S_{main}=S_{main}^{+}\cup S_{main}^{-}$$
where $S_{main}^{+}$ and $S_{main}^{-}$ represents peptide sequences of anti-angiogenic and non-antiangiogenic classes, respectively, from the $S_{main}$ dataset while the symbol ∪ represents the union from the set theory. Based on the study of Ramaprasad et al. [27], AntiAngioPred was developed by using the NT15 dataset, which provided the highest prediction accuracy thus far. Therefore, to make a fair comparison with this method, the N-terminus dataset was considered. In this respect, the NT15 dataset containing the first 15 residues from the N-terminus region of peptide was used in our comparative investigation. After preparing the NT15 dataset, it consisted of 99 anti-angiogenic and 101 non-antiangiogenic peptides. The NT15 dataset ($S_{NT15}$) used in this study can be formulated as: (2)$$S_{NT15}=S_{NT15}^{+}\cup S_{NT15}^{-}$$
where $S_{NT15}^{+}$ and $S_{NT15}^{-}$ represents the peptide sequences of anti-angiogenic and non-antiangiogenic classes, respectively, from the $S_{NT15}$ dataset. Moreover, the $S_{main}$ and $S_{NT15}$ datasets were randomly divided into training (for cross-validation test) and testing (for independent validation test) sets, where 80% and 20% of the two datasets were used as training and testing sets. A summary of the size distribution of the training and testing sets is provided in Table 7.

### 3.2. Feature Representation

In order to develop a robust and interpretable sequence-based computational model, the critical process is to represent the peptides in such a way so as to afford a comprehensive and proper description of the feature that could well reflect their functions. Amongst the various types of sequence features that are available, easy and interpretable features are those pertaining to the amino acid composition (AAC), dipeptide composition (DPC), and physicochemical properties (PCP). 

AAC and DPC are the proportions of each amino acid and dipeptide in a peptide sequence **P** that are expressed as fixed lengths of 20 and 400, respectively. Thus, in terms of AAC and DPC features, a peptide **P** can be expressed by vectors with 20D and 400D (dimension) spaces, respectively, as formulated by: (3)$$Ρ={\left[{\mathrm{aa}}_{1},{\mathrm{aa}}_{2},\ldots ,\;{\mathrm{aa}}_{20}\right]}^{T}$$
(4)$$Ρ={\left[{\mathrm{dp}}_{1},{\mathrm{dp}}_{2},\ldots ,\;{\mathrm{dp}}_{400}\right]}^{T}$$
where T is the transposed operator, while aa_1_, aa_2_, …, aa_20_ and dp_1_, dp_2_, …, dp_400_ are occurrence frequencies of the 20 and 400 native amino acids and dipeptides, respectively, in a peptide sequence **P**. 

PCP is one of the most intuitive features associated with biophysical and biochemical reactions. In fact, a total of 531 PCPs without NA values were derived from version 9.0 of the Amino acid index database (AAindex) [58], which is a collection of the published literature pertaining to different physicochemical and biophysical properties of amino acids and pairs of amino acids (http://www.genome.jp/aaindex/). Each physicochemical property consisted of a set of 20 numerical values for amino acids. The PCP feature has been extensively used for the prediction and analysis of various protein [53,55,80] and peptide [56,57] functions. To utilize PCP features for extracting a peptide sequence, peptide with the length of L amino acid residues is encoded into an L-dimensional vector of 531 PCPs (531D). 

As mentioned in previous studies [81,82] and shown in Equations (3) and (4), AAC and DPC features only provide the compositional information of a peptide sequence, but all the sequence-order information may be completely lost. To remedy this limitation, the pseudo amino acid composition (PseAAC) and amphiphilic pseudo amino acid composition (Am-PseAAC) approaches were proposed by Chou [81,82]. According to Chou’s PseAAC, the general form of PseAAC for a peptide **P** is formulated by:(5)$$Ρ={\left[\Psi_{1},\Psi_{2},\ldots ,\Psi_{u},\ldots ,\;\Psi_{\Omega }\right]}^{T}$$
where the subscript Ω is an integer to reflect the feature’s dimension. The value of Ω and the component of $\Psi_{u}$, where u=1,2,…,Ω is dependent on the protein or peptide sequences. In this study, the parameters of PseAAC (i.e., the discrete correlation factor λ and weight of the sequence information ϖ) were estimated by using the optimization procedure as described hereafter. The dimension of PseAAC feature is 20 + λ×ϖ. Since the hydrophobic and hydrophilic properties of proteins play an important role in the folding and interaction of proteins, Am-PseAAC was introduced by Chou [82]. The dimension of Am-PseAAC feature is 20 + 2λ. The first 20 components are the 20 basic AAC ($p_{1},p_{2},\ldots ,\;p_{20}$) while the next 2λ ones denote the set of correlation factors that reveal the physicochemical properties such as hydrophobicity and hydrophilicity along a protein or peptide sequence as formulated by:(6)$$Ρ={\left[p_{1},p_{2},\ldots ,\;p_{20},p_{20+\lambda },p_{20+\lambda +1},\ldots p_{20+2\lambda }\right]}^{T}$$

In this study, the five aforementioned features of peptide sequences were generated by using the protr package in the R programming environment [83]. The parameters of PseAAC (weight^1^ and lamda^1^) and Am-PseAAC (weight^2^ and lamda^2^) were optimized by varying weight and lambda values from 0 to 1 and 1 to 10 with step sizes of 0.1 and 1, respectively, on the whole $S_{main}$ and $S_{NT15}$ datasets as assessed by a 5-fold CV procedure. 

### 3.3. Random Forest

The learning classifier employed herein was constructed using the original RF algorithm [84,85]. This model is an ensemble model consisting of many classification and regression tree (CART) classifiers that improves prediction performances of CART classifiers by growing a number of weak CART classifiers. Prediction results from the classification task is obtained by using simple voting from amongst outputs all trees to arrive at one final prediction. In regression, a final prediction is the average of prediction results from many trees. In order to construct the RF model, each tree is built as follows: (i) a bootstrap sample, which is used as a training set for the current tree, is obtained from the whole training set consisting of N peptides. In the meanwhile, peptides which are not used for constructing the current tree are place in an out-of-bag (OOB) set, where the size of the OOB set is around N/3, (ii) the m selected features from the whole M features is derived from the best split by CART model, (iii) each tree is grown to the largest possible extent, (iv) if there is no pruning then the procedure is terminated. Herein, the RF classifier was established using the randomForest package in the R software [84]. To enhance the performance of the RF model, two parameters including ntree (i.e., the number of trees used for constructing the RF classifier) and mtry (i.e., the number of random candidate features) were determined using the caret R package [86] with the 5-fold CV approach. The search space of ntree and mtry are {100,200,…,500} and {1,2,…,10} with the steps of 100 and 1, respectively. Previously, RF model has been successfully used in the prediction of various functions and properties of peptides and proteins [55,56,57,87,88] as well as other biological or chemical entities [89,90,91,92,93].

### 3.4. Identification of Important Features

In this work, we performed the analysis and identification of feature importance for each type of sequence feature by using the RF method to provide a better understanding of the biophysical and biochemical properties of anti-angiogenic activities of peptides. In RF method, the OOB approach is used for evaluating the feature importance as follows: (i) two-thirds of the training data is utilized to construct the predictive classifier while the remaining is used for evaluating the performance of such classifier and (ii) the feature importance of each feature can be evaluated by measuring the decrease of the prediction performance. It should be noted that the performance evaluation of the model can be either accuracy or Gini index. In summary, the RF method provides two measures for ranking feature importance, i.e., the mean decrease of Gini index (MDGI) and the mean decrease of prediction accuracy. Since Calle and Urrea [94] demonstrated that the MDGI provided a more robust result as compared to the mean decrease of prediction accuracy, we utilized the MDGI value to rank the importance of interpretable features including AAC, DPC, and PCP. Until now, these three features have been used to characterize many peptides and proteins, such as predicting HIV-1 CRF01-AE co-receptor usage [95], predicting protein crystallization [53,96], predicting the oligomeric states of fluorescent proteins [88], predicting the bioactivity of host defense peptides [87], prediction of human leukocyte antigen gene [80,97], predicting antifreeze proteins [55], predicting the hemolytic activity of peptides [56], and predicting antihypertensive activity of peptides [57]. 

The Gini index can be defined as MDGI is an impurity measure that corresponds to the ability of each feature in discriminating the sample classes. The Gini index can be defined as
(7)$$1-\sum_{c=1}^{2}p^{2}(c|t)$$
where *p*^2^(*c*|*t*) denotes the estimated class probability for node t in a tree classifier and c in the class label (i.e., either anti-angiogenic or non-antiangiogenic peptides). In order to increase the reliability for identifying the feature importance, 10 RF models were constructed by varying the mtry parameter settings from 2 to 20 (mtry = 2, 3, 5, 7, 9, 11, 13, 15, 17, 20) and fixing the ntree parameter with 100 [55,56,57]. Finally, the average value of MDGI on 10 runs of feature importance estimations were used in this study. Features with the largest MDGI value is considered to be an important feature as it significantly contributes to the prediction performance. Herein, the MDGI values of feature importance for each type of sequence feature was estimated using the randomForest package in the R software [84].

### 3.5. Performance Evaluation

In a statistical prediction, the following three testing methods are often used to evaluate the prediction performance in practical applications: sub-sampling test or k-fold cross-validation (k-fold CV), jackknife test, and independent validation test or external test. The sub-sampling and jackknife test are popular cross-validation methods to assess the predictive capability of the model. Meanwhile, the external test is considered as one of the most rigorous and reliable methods for the cross-validation purposes in statistics. In k-fold cross-validation procedure, the training set is randomly separated into k subsets. From the k subsets, a single subset is taken as the testing set to validate the prediction model that is trained and learned by the remaining k-1 subsets. This process is repeated k times, until each subset had been used as the testing set. During the jackknifing process, a single sample in the whole dataset having N samples is taken as the testing set and the remaining N-1 samples are used for training the model. This process is repeated N times, until each ssample had been used as the testing set. 

In order to evaluate the prediction ability of the model, the following sets of four metrics are used as follows: (8)$$\mathrm{Ac}=\frac{\mathrm{TP}+\mathrm{TN}}{\left(\mathrm{TP}+\mathrm{TN}+\mathrm{FP}+\mathrm{FN}\right)}$$
(9)$$\mathrm{Sn}=\frac{\mathrm{TP}}{\left(\mathrm{TP}+\mathrm{FN}\right)}$$
(10)$$\mathrm{Sp}=\frac{\mathrm{TN}}{\left(\mathrm{TN}+\mathrm{FP}\right)}$$
(11)$$\mathrm{MCC}=\frac{\mathrm{TP}\times \mathrm{TN}-\mathrm{FP}\times \mathrm{FN}}{\sqrt{\left(\mathrm{TP}+\mathrm{FP}\right)\left(\mathrm{TP}+\mathrm{FN}\right)\left(\mathrm{TN}+\mathrm{FP}\right)\left(\mathrm{TN}+\mathrm{FN}\right)}}$$
where Ac, Sn, Sp, and MCC are called accuracy, sensitivity, specificity, and Matthews correlation coefficient, respectively. TP, TN, FP, and FN represent the instances of true positive, true negative, false positive, and false negative, respectively. Moreover, in order to evaluate the prediction performance of models using threshold-independent parameters, the receiver operating characteristic (ROC) curves were plotted by the pROC package in the R software [98]. The area under the ROC curve (auAUC) was used to measure the prediction performance, where AUC values of 0.5 and 1 are indicative of perfect and random models, respectively.

### 3.6. Reproducible Research

To ensure the reproducibility of the models proposed herein, all R codes and datasets used in the construction of the predictive models, graphical figures and the TargetAntiAngio web server are available on GitHub at https://github.com/Shoombuatong2527/targetantiangio (accessed on: 1 April 2019) and https://github.com/chaninlab/targetantiangio-webserver (accessed on: 1 April 2019). 

## 4. Conclusions

Anti-angiogenesis plays a fundamental role in tumor growth, invasion, and metastatic dissemination. Several anti-angiogenic peptides have been developed in order to promote effective cancer treatment as well as enhanced survival rate. Therefore, computational methods that can predict and analyze anti-angiogenic peptides based on peptide sequences are highly desirable. In this study, we have developed a new computational model named TargetAntiAngio for predicting and analyzing anti-angiogenic peptides based on sequence information. TargetAntiAngio is developed using the random forest classifier in conjunction with a combination of amino acid composition, pseudo amino acid composition and amphiphilic pseudo amino acid composition. The prediction results for both cross-validation and independent validation tests on the benchmark dataset demonstrated that TargetAntiAngio can pick out informative features as well as improve prediction performances. In addition, a thorough analysis of the peptide feature importance was conducted to unravel and rationalize the biophysical and biochemical properties of anti-angiogenic activities of peptides. Finally, to help potential users of TargetAntiAngio, a web-server based on the optimal model has been established and made freely available online at http://codes.bio/targetantiangio/ thereby allowing users easy access to their desired results.

## Figures and Tables

**Figure 1 ijms-20-02950-f001:**
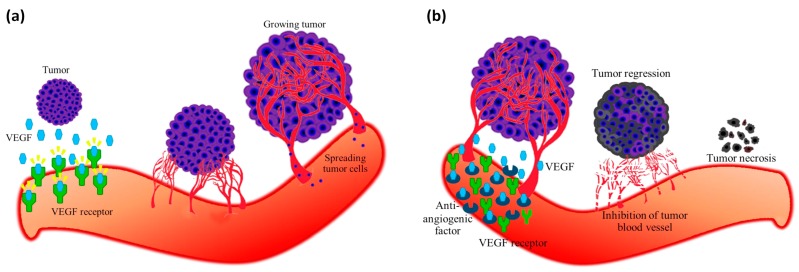
Angiogenesis is regulated by a local equilibrium between pro-angiogenic such as vascular endothelial growth factor (VEGF), platelet-derived endothelial growth factor (PDGF), fibroblast growth factor (FGF), and angiopoietins and anti-angiogenic molecules such as endostatin, PF4 and TSP-1. It is switched on when tumor cells require oxygen and nutrients. Tumor cells produce VEGF and then secretes them into surrounding tissues. When VEGF binds to its receptor on the outer surface of endothelial cells, it activates endothelial cells that subsequently drives the development of new blood vessels from pre-existing vasculatures. Blood vessels gradually grow and expand to tumor cells whereby tumor cells continuously proliferate and spread into the blood circulation. Cancer progression is induced by an overexpression of pro-angiogenic factors (**a**). Disruption of the vascular supply can be mediated by blocking pro-angiogenic factors or via the use of anti-angiogenic factors as therapeutic drug is anticipated to increase the survival rate of cancer patients. Anti-angiogenic factor binds to VEGF thereby leading to the inhibition of neovascularization and tumor growth thereby leading to a decrease of metastasis. Eventually, tumor cells which are devoid of fuels (e.g., oxygen and nutrients) gently regress and become tumor necrosis (**b**).

**Figure 2 ijms-20-02950-f002:**
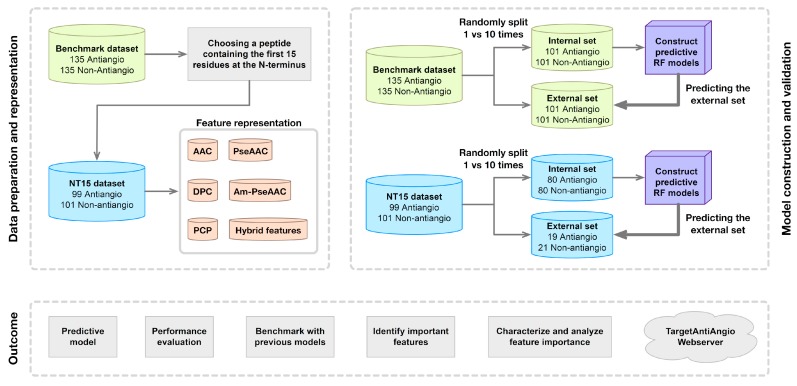
Schematic framework of TargetAntiAngio.

**Figure 3 ijms-20-02950-f003:**
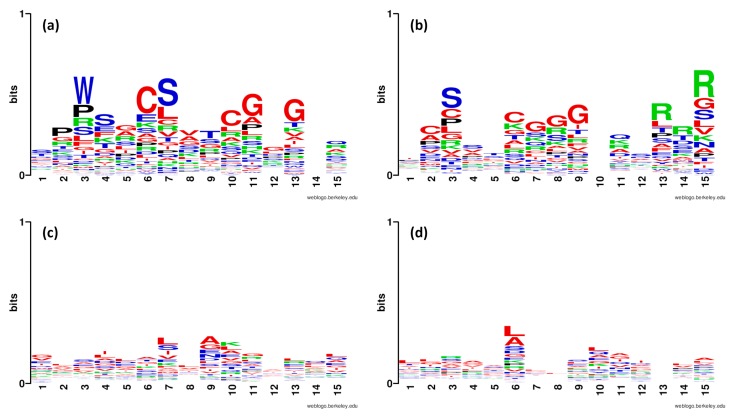
Sequence logo representations of antiangiogenic and non-antiangiogenic peptides. Shown are the sequence logo of the first and last 15 residues at N- and C-terminal regions from antiangiogenic peptides (**a**,**b**) and non-antiangiogenic peptides (**c**,**d**).

**Figure 4 ijms-20-02950-f004:**
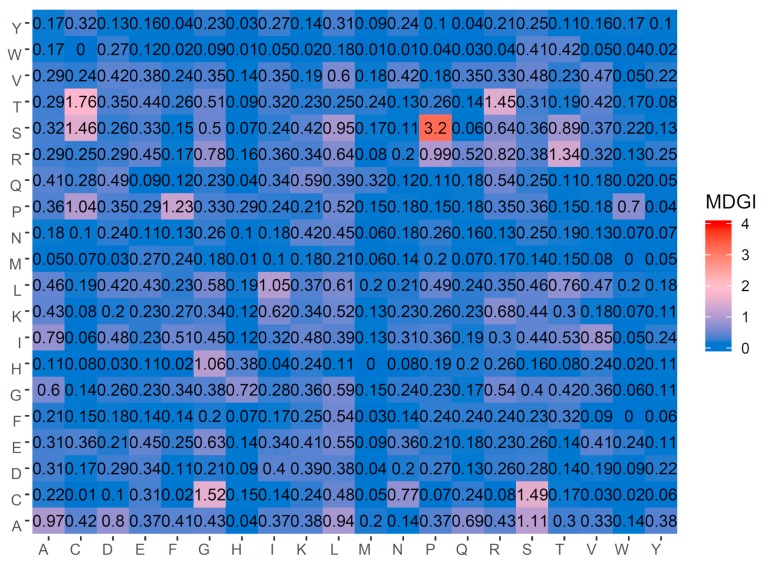
Heat map of the mean decrease of Gini index (MDGI) of dipeptide compositions. It should be noted that features with the largest value of MDGI are the most important.

**Figure 5 ijms-20-02950-f005:**
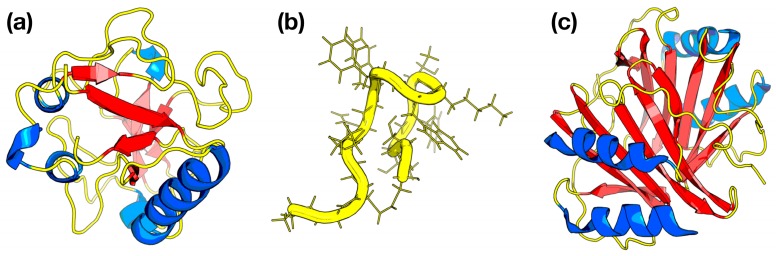
Three-dimensional structures of established anti-angiogenic inhibitors consisting of endostatin (PDB id 1KOE) (**a**), somatostatin (PDB id 2MI1) (**b**), and Platelet factor-4 (PDB id 1RHP) (**c**). α-helix, β-sheet, and loop are shown in blue, red and yellow colors, respectively.

**Figure 6 ijms-20-02950-f006:**
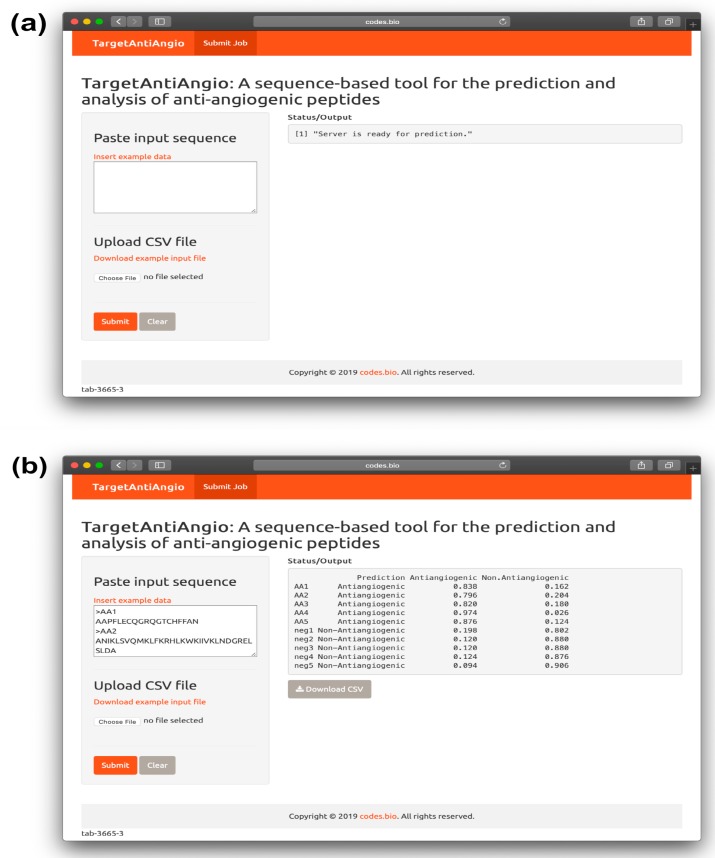
Screenshot of the TargetAntiAngio web server before (**a**) and after (**b**) submission of the input query sequence.

**Table 1 ijms-20-02950-t001:** Summary of existing methods for predicting anti-angiogenic peptides.

Method	Classifier ^a^	Sequence Feature (No. of Feature Used) ^b^	Independent Test	Web Server
AntiAngioPred [27]	SVM	AAC (20)	Yes	Yes
Blanco et al.’s method [28]	glmnet	AAC, DPC, TC (200)	No	No
AntAngioCOOL [29]	PART	PseAAC, *k*-mer composition, RAAC, PCP, AC (2,343)	No	No
TargetAntiAngio (this study)	RF	AAC, PseAAC, Am-PseAAC (48)	Yes	Yes

^a^ glmnet: a generalized linear model, PART: recursive partitioning for classification, regression and survival trees, RF: random forest, SVM: support vector machine. ^b^ AAC: amino acid composition, AC: atomic profile, Am-PseAAC: amphiphilic pseudo amino acid composition, DPC: dipeptide composition, PCP: physicochemical properties, PseACC: pseudo amino acid composition, RACC: reduce amino acid composition, TC: tripeptide composition. The method is assessed by an independent validation test with *N* rounds of random splits.

**Table 2 ijms-20-02950-t002:** Performance comparison of RF models built with various types of sequence features. Models were evaluated by means of five-fold cross-validation and independent validation test using benchmark and NT15 datasets subjected to one round of random split.

Feature	Dataset	5-Fold CV			Independent Test				
		Ac (%)	MCC	auROC	Ac (%)	Sn (%)	Sp (%)	MCC	auROC
ACC	Benckmark	71.03	0.42	0.81	72.12	67.86	76.92	0.45	0.77
	NT15	75.00	0.50	0.80	77.50	90.48	63.16	0.56	0.82
PseAAC	Benckmark	73.83	0.48	0.78	72.22	85.71	57.69	0.45	0.81
	NT15	73.75	0.48	0.80	72.50	85.71	57.90	0.46	0.83
Am-PseAAC	Benckmark	71.96	0.44	0.76	72.22	82.14	61.54	0.45	0.76
	NT15	72.50	0.45	0.80	75.00	76.19	73.68	0.50	0.80
DPC	Benckmark	68.22	0.37	0.75	70.37	82.14	57.69	0.41	0.72
	NT15	72.50	0.45	0.79	72.50	95.24	47.37	0.49	0.69
PCP	Benckmark	60.75	0.22	0.67	61.11	67.86	53.85	0.22	0.65
	NT15	67.50	0.36	0.72	67.50	76.19	57.90	0.35	0.74
AAC+PseAAC	Benckmark	72.43	0.45	0.79	75.93	85.71	65.39	0.52	0.80
	NT15	74.38	0.50	0.77	77.00	85.71	68.42	0.55	0.83
AAC+Am-PseAAC	Benckmark	70.09	0.41	0.76	74.07	89.29	57.69	0.50	0.83
	NT15	74.38	0.50	0.81	75.00	71.43	78.95	0.50	0.79
PseAAC+Am-PseAAC	Benckmark	72.90	0.46	0.77	77.78	82.14	73.08	0.56	0.83
	NT15	75.00	0.50	0.82	75.00	85.71	63.16	0.50	0.85
AAC+PseAAC+Am-PseAAC	Benckmark	71.03	0.42	0.78	74.07	75.00	73.08	0.48	0.82
	NT15	75.00	0.50	0.82	77.50	90.48	63.16	0.56	0.84

Parameters of PseAAC (weight^1^ and lamda^1^) and Am-PseAAC (weight^2^ and lamda^2^) were optimized by varying their values and assessed by a 5-fold CV procedure. Values of weight^1^, weight^2^, lamda^1^, and lamda^2^ as performed on the benchmark and NT15 datasets are (0.9, 0.9, 1, and 1) and (0.1, 0.2, 2, and 3), respectively.

**Table 3 ijms-20-02950-t003:** Performance comparison of RF models built with various types of sequence features. Models were evaluated by means of five-fold cross-validation and independent validation test using benchmark and NT15 datasets subjected to ten rounds of random splits.

Feature	Dataset	5-Fold CV			Independent Test				
		Ac (%)	MCC	auROC	Ac (%)	Sn (%)	Sp (%)	MCC	auROC
ACC	Benckmark	70.84 ± 1.54	0.42 ± 0.03	0.79 ± 0.01	73.33 ± 1.01	77.14 ± 8.60	69.23 ± 9.81	0.47 ± 0.02	0.79 ± 0.02
	NT15	74.12 ± 2.10	0.49 ± 0.04	0.80 ± 0.02	77.00 ± 2.09	84.76 ± 6.21	68.42 ± 8.32	0.55 ± 0.04	0.82 ± 0.02
PseAAC	Benckmark	71.78 ± 2.13	0.44 ± 0.04	0.77 ± 0.01	72.96 ± 1.66	80.00 ± 4.07	65.38 ± 6.08	0.46 ± 0.03	0.81 ± 0.02
	NT15	71.62 ± 1.44	0.43 ± 0.03	0.78 ± 0.02	73.50 ± 1.37	80.95 ± 8.25	65.26 ± 11.53	0.48 ± 0.03	0.81 ± 0.03
Am-PseAAC	Benckmark	70.47 ± 2.10	0.41 ± 0.04	0.75 ± 0.02	72.96 ± 2.11	75.71 ± 9.58	70.00 ± 8.34	0.46 ± 0.05	0.79 ± 0.04
	NT15	72.38 ± 2.31	0.45 ± 0.05	0.81 ± 0.01	73.50 ± 1.37	76.19 ± 12.14	70.53 ± 12.68	0.48 ± 0.02	0.79 ± 0.04
DPC	Benckmark	68.32 ± 0.84	0.37 ± 0.02	0.74 ± 0.01	69.63 ± 2.11	78.57 ± 9.45	60.00 ± 5.83	0.40 ± 0.05	0.74 ± 0.02
	NT15	71.50 ± 2.01	0.43 ± 0.04	0.78 ± 0.02	69.50 ± 3.26	78.09 ± 10.43	60.00 ± 7.98	0.40 ± 0.08	0.75 ± 0.07
PCP	Benckmark	60.19 ± 2.44	0.20 ± 0.05	0.65 ± 0.02	61.85 ± 2.11	62.14 ± 10.59	61.54 ± 9.81	0.24 ± 0.04	0.66 ± 0.03
	NT15	68.00 ± 1.03	0.36 ± 0.02	0.74 ± 0.02	67.50 ± 2.50	72.38 ± 7.82	62.11 ± 11.41	0.35 ± 0.05	0.72 ± 0.07
AAC+PseAAC	Benckmark	72.24 ± 0.53	0.45 ± 0.01	0.79 ± 0.01	74.81 ± 1.01	81.43 ± 4.66	67.69 ± 5.83	0.50 ± 0.02	0.81 ± 0.04
	NT15	73.50 ± 1.44	0.47 ± 0.03	0.79 ± 0.02	76.50 ± 1.37	84.76 ± 7.82	67.37 ± 10.79	0.54 ± 0.02	0.82 ± 0.05
AAC+Am-PseAAC	Benckmark	70.37 ± 1.13	0.41 ± 0.02	0.77 ± 0.02	73.33 ± 1.01	85.00 ± 4.66	60.77 ± 5.70	0.48 ± 0.02	0.78 ± 0.04
	NT15	73.00 ± 0.81	0.47 ± 0.02	0.80 ± 0.02	75.50 ± 1.12	83.81 ± 7.97	66.32 ± 9.56	0.52 ± 0.02	0.82 ± 0.06
PseAAC+Am-PseAAC	Benckmark	73.18 ± 1.57	0.47 ± 0.03	0.78 ± 0.01	73.33 ± 2.81	80.71 ± 1.96	65.38 ± 5.44	0.47 ± 0.05	0.78 ± 0.05
	NT15	73.88 ± 2.14	0.48 ± 0.04	0.80 ± 0.02	75.00 ± 1.77	80.95 ± 5.83	68.42 ± 6.45	0.50 ± 0.04	0.80 ± 0.05
AAC+PseAAC+Am-PseAAC	Benckmark	70.37 ± 1.22	0.41 ± 0.02	0.77 ± 0.02	74.07 ± 1.31	82.14 ± 5.05	65.38 ± 7.20	0.49 ± 0.02	0.81 ± 0.01
	NT15	74.62 ± 1.57	0.50 ± 0.03	0.81 ± 0.01	77.50 ± 1.77	84.76 ± 10.32	69.47 ± 8.65	0.56 ± 0.03	0.83 ± 0.03

Parameters of PseAAC (weight^1^ and lamda^1^) and Am-PseAAC (weight^2^ and lamda^2^) were optimized by varying their values and assessed by a 5-fold CV procedure. Values of weight^1^, weight^2^, lamda^1^, and lamda^2^ as performed on the benchmark and NT15 datasets are (0.9, 0.9, 1, and 1) and (0.1, 0.2, 2, and 3), respectively.

**Table 4 ijms-20-02950-t004:** Performance comparisons between TargetAntiAngio and AntiAngioPred assessed by 5-fold cross-validation and independent validation tests on NT15 dataset.

Sampling Time	Method	Cross-Validation		Independent Test			
		Ac (%)	MCC	Ac (%)	Sn (%)	Sp (%)	MCC
1 round	AntiAngioPred ^a^	80.90	0.62	75.00	-	-	0.51
	TargetAntiAngio	75.00	0.50	77.50	90.48	63.16	0.56
N rounds ^b^	AntiAngioPred ^a^	-	-	74.96	72.90	76.80	0.50
	TargetAntiAngio	74.62	0.50	77.50	84.76	69.47	0.56

^a^ Results were reported from the work of AntiAngioPred. ^b^ N represents the number of 5 and 10 rounds of random splits for performing the prediction results of AntiAngioPred and TargetAntiAngio, respectively.

**Table 5 ijms-20-02950-t005:** Amino acid compositions (%) of antiangiogenic (Angio) and non-antiangiogenic (non-Angio) peptides along with their mean decrease of Gini index (MDGI) values.

Amino acid	Anti-Angio (%)	Non-Anti-Angio (%)	Difference	p-value	MDGI
A-Ala	0.053	0.086	−0.033 (20)	<0.05	9.21 (4)
C-Cys	0.047	0.014	0.034 (2)	<0.05	15.90 (1)
D-Asp	0.047	0.052	−0.005 (13)	0.568	5.02 (12)
E-Glu	0.046	0.065	−0.019 (17)	<0.05	6.68 (7)
F-Phe	0.030	0.043	−0.013 (15)	<0.05	4.89 (13)
G-Gly	0.081	0.073	0.008 (7)	0.420	5.18 (11)
H-His	0.030	0.024	0.007 (8)	0.373	4.58 (14)
I-Ile	0.046	0.064	−0.017 (16)	<0.05	5.26 (10)
K-Lys	0.056	0.056	0.001 (9)	0.933	6.59 (8)
L-Leu	0.067	0.095	−0.028 (19)	<0.05	8.41 (5)
M-Met	0.019	0.023	−0.004 (12)	0.366	3.45 (20)
N-Asn	0.037	0.040	−0.003 (11)	0.657	3.71 (18)
P-Pro	0.060	0.045	0.016 (4)	<0.05	6.40 (9)
Q-Gln	0.039	0.042	−0.002 (10)	0.757	4.47 (15)
R-Arg	0.088	0.055	0.032 (3)	<0.05	8.31 (6)
S-Ser	0.096	0.057	0.039 (1)	<0.05	14.43 (2)
T-Thr	0.062	0.054	0.008 (6)	0.232	3.77 (17)
V-Val	0.048	0.073	−0.025 (18)	<0.05	9.58 (3)
W-Trp	0.023	0.012	0.012 (5)	<0.05	3.95 (16)
Y-Tyr	0.023	0.029	−0.007 (14)	0.210	3.45 (19)

**Table 6 ijms-20-02950-t006:** Ten top-ranked physicochemical properties from the AAindex having the highest MDGI values.

Rank	AAindex	MDGI	Description
1	CHOP780216	0.73	Normalized frequency of the 2nd and 3rd residues in turn (Chou-Fasman, 1978b)
2	CHOP780215	0.61	Frequency of the 4th residue in turn (Chou-Fasman, 1978b)
3	MIYS990104	0.58	Optimized relative partition energies—method C (Miyazawa-Jernigan, 1999)
4	CHOP780214	0.54	Frequency of the 3rd residue in turn (Chou-Fasman, 1978b)
5	ENGD860101	0.54	Hydrophobicity index (Engelman et al., 1986)
6	OLSK800101	0.53	Average internal preferences (Olsen, 1980)
7	MIYS990105	0.53	Optimized relative partition energies—method D (Miyazawa-Jernigan, 1999)
8	LEVM780104	0.52	Normalized frequency of alpha-helix, unweighted (Levitt, 1978)
9	MIYS990101	0.52	Relative partition energies derived by the Bethe approximation (Miyazawa-Jernigan, 1999)
10	KIDA850101	0.50	Hydrophobicity-related index (Kidera et al., 1985)

MDGI: Mean decrease of Gini index.

**Table 7 ijms-20-02950-t007:** Summary of two datasets for evaluating the predictors of anti-angiogenic peptides as obtained from Ramaprasad et al.

Dataset	$S_{main}$		$S_{NT15}$	
	Anti-angio	Non-anti-angio	Anti-angio	Non-anti-angio
Original data	137	137	99	101
Training set	101	101	80	80
Testing set	36	36	19	21

Anti-angio and non-anti-angio represent anti-angiogenic and non-antiangiogenic peptides, respectively.

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
