# Peer review of "TargetAntiAngio: A Sequence-Based Tool for the Prediction and Analysis of Anti-Angiogenic Peptides"

_ijms, 2019, doi:10.3390/ijms20122950_

Round 1

Reviewer 1 Report

The authors present a computational tool, dubbed TargetAntiAngio, which aims at predicting anti-angiogenic peptides. The concept is not strictly novel, while the results and yields range between good and excellent. The tool seems robust and relatively simple from the conceptual perspective, and may be of interest to scientists working in the field and related areas.

Author Response

Point 1: The authors present a computational tool, dubbed TargetAntiAngio, which aims at predicting anti-angiogenic peptides. The concept is not strictly novel, while the results and yields range between good and excellent. The tool seems robust and relatively simple from the conceptual perspective, and may be of interest to scientists working in the field and related areas.

Response 1: Thank you for the encouraging and kind words of inspiration

Reviewer 2 Report

Specific questions:

In this manuscript, the authors would like to establish a novel hypothesis to target angiogenesis by identifying novel peptides using computational model. The authors suggest that using this model system the anti-angiognic peptides with Cys residues play an important role which is located in C-terminal domain contribute to the inhibition of angiogenesis and cell migration.  The authors provided substantial evidence suggesting the proposed cooperation of Cys residue at the C-terminal end, however, the direct interaction in vivo must be shown.

The authors stated that “Considering that the independent validation test with more than 1 rounds of random splits is the most rigorous cross-validation method, it can be claimed that TargetAntiAngio is a more efficient and reliable approach for predicting anti-angiogenic peptides.” This is a very strong statement. Without using mutated Cys regulatory element it is hard to establish.

Minor comments:

Fig.4 and Fig 5 legends can be elaborated for quick gain of understanding.

Fig 6 legend is not clear. It is repeated with Fig5 legend, need to be fixed.

As the title of the section Is Results and discussion, the title of each results can be given in one line conclusion rather that just one word headings. 

Author Response

Response to Reviewer 2 Comments

Point 1: The authors suggest that using this model system the anti-angiognic peptides with Cys residues play an important role which is located in C-terminal domain contribute to the inhibition of angiogenesis and cell migration.  The authors provided substantial evidence suggesting the proposed cooperation of Cys residue at the C-terminal end, however, the direct interaction in vivo must be shown.

Response 1: Thank you for the kind suggestion. As suggested, the direction interaction information was added at pages #11, #12 and #15 of the revised manuscript.

Point 2: The authors stated that “Considering that the independent validation test with more than 1 rounds of random splits is the most rigorous cross-validation method, it can be claimed that TargetAntiAngio is a more efficient and reliable approach for predicting anti-angiogenic peptides.” This is a very strong statement.

Response 2: Thank you for the kind suggestion. As suggested, this statement was deleted from the manuscript.

Point 3: Without using mutated Cys regulatory element it is hard to establish.

Response 3: Thank you for kind suggestion. We have already added this point at page #12 of the revised manuscript.

Point 4: Fig.4 and Fig 5 legends can be elaborated for quick gain of understanding.

Response 4: Thank you for the comment, correction has been made as suggested.

-                      The figure legend of Fig 4 (page #11) was corrected to include an additional statement about how to interpret the significant feature from the heatmap as follows: “It should be noted that features with the largest value of MDGI are the most important.”

-                      The figure legend of Fig 5 (page #14) was modified to include the phrase “well-known” as these peptides are the well established anti-angiogenic inhibitors. Further information about these three peptides are already mentioned in the manuscript in section 2.4. Mechanistic interpretation of informative PCP.

Point 5: Fig 6 legend is not clear. It is repeated with Fig5 legend, need to be fixed.

Response 5: Thank you for the kind suggestion. This point has been corrected at page #16 of the revised manuscript.

Point 6: As the title of the section Is Results and discussion, the title of each results can be given in one line conclusion rather that just one word headings.

Response 6: Thank you for the kind suggestion. We have gone through the text one more time to check the Results heading such that they are explanatory in a couple of words.